# Facile Synthesis of 3-Substituted Thiazolo[2,3-α]tetrahydroisoquinolines

**DOI:** 10.3390/molecules26206126

**Published:** 2021-10-11

**Authors:** Sheng-Han Huang, Wan-Yu Huang, Guo-Lun Zhang, Te-Fang Yang

**Affiliations:** Department of Applied Chemistry, National Chi Nan University, Puli, Nantou 545, Taiwan; s99324901@ncnu.edu.tw (S.-H.H.); redvedaa@gmail.com (W.-Y.H.); a26332915@gmail.com (G.-L.Z.)

**Keywords:** isoquinoline, aza-Michael addition, heterocycles

## Abstract

It was found that 4-hydroxy-2-butenoic ester **(11)** could not react with 3,4-dihydro-isoquinoline (**4a**). Individual addition reactions of γ-mercapto-α,β-unsaturated esters (**18)** and -unsaturated amide (**19)** with 3,4-dihydroisoquinolines (**4**) were carried out under appropriate conditions to provide the corresponding thiazolo[2,3-α]isoquinoline derivatives with good yields (up to 87%) and significant diastereomeric selectivity. The mechanism of the crucial reaction was discussed.

## 1. Introduction

Alkaloids salsolidine, an MAOA inhibitor, and tetrahydropapaverine could be synthesized using dihydrothiazolo[2,3-α]isoquinolinones as the precursors and/or intermediates [1,2,3,4]. However, little attention was paid to the synthesis of thiazolo[2,3-α]tetrahydroisoquinolines, although they are the analogs of oxazolidine moiety contained in the molecular structure of some potential antitumors, such as quinocarcin (**1**) and tetrazomine (**2**) (Figure 1) [5,6,7,8,9]. Very recently, we reported that oxazolo[2,3-α]tetrahydroisoquinolines **3**, an analog of oxazolidine structure in quinocarcin and tetrazomine, could be synthesized via the reaction of 3,4-dihydroisoquinolines (**4**) with γ-hydroxy-α,β-unsaturated ketones **5** at room temperature (Figure 1) [10,11]. Thus, for the generality and application of this tandem reaction, which was carried out under green conditions, we tried to prepare γ-mercapto-α,β-unsaturated ketone **8** and expected that it might undergo addition reaction with **4a** to furnish thiazolo[2,3-α]tetrahydroisoquinolines **10** (Figure 2). Unfortunately, since the reaction of ylide **6** with diol **7** afforded dimer **9a** or **9b** instead of ketone **8** [12], there was no opportunity of running the corresponding reaction of isoquinolines **4** for the synthesis of structure **10**. Furthermore, it is noteworthy that 4-hydroxy-2-butenoic ester **11** could not react with 3,4-dihydroisoquinoline (**4a**) to provide structure **12** (Figure 3), either. This phenomenon might be due to the strong electron-donation effect of–OR_2_ on the carbonyl part of the ester moiety in **11**, such that this compound was not versatile enough for the aza-Michael reaction with **4a**.

To the best of our knowledge, to date, there are only a few examples of the synthesis of thiazolo[2,3-α]isoquinoline published in the literature. For example, treatment of 3,4-dihydroisoquinoline **13** [10,11], with ethylene sulfide could provide compounds **14** (Figure 4) [3]. On the other hand, in 1975, Bradsher and co-workers reported that parent thiazolo[2,3-α]tetrahydroisoquinoline (**17**) could be obtained starting with a cyclization of isoquinolinium salt **15**, followed by a disproportionation reaction [13]. Nevertheless, we found that the individual reaction of γ-mercapto-α,β-unsaturated esters **18** and -unsaturated amide **19** with 3,4-dihydroisoquinolines (**4**) could be carried out under green conditions to afford the corresponding thiazolo[2,3-α]isoquinolines **20** and **21** [14] (Figure 4).

## 2. Results and Discussion

Some reactants for the synthesis of the title compounds were prepared in terms of the procedures published in the literature. As shown in Figure 5, (Appendix A) compounds **18a**–**18c** could be obtained from the individual reactions of **22a**–**22c** with **7** [14]. On the other hand, reaction of **23** with **24** furnished γ-mercapto-α,β-unsaturated amide **19** [15] (Figure 6).

Thus, in order to efficiently obtain the title compounds, the addition reaction conditions were screened, and the results are listed in Table 1. The 3,4-dihydroisoquinoline (**4a**, R=H) and compound **18a** were selected as the reactant of the model reaction. At first, THF was adopted as the solvent for the reaction (entries 1~5) in the presence of either weak base or acid at room temperature. However, the target product was obtained in moderate yields. Then, it was found that excess amount of acetic acid could be an efficient additive for the reaction in THF (entry 6). Furthermore, the yield of **20aa** increased to 70% in ethyl acetate (entry 8), which is a less toxic solvent and easier to handle than THF. Thus, it was expected that EtOAc could be one of the good solvents for the reaction.

Nevertheless, the yield of the product was not improved in some other polar solvents (entries 9~11 and 13). Finally, the reaction afforded the target molecule in good yield in hot ethyl acetate and acetic acid (entries 14 and 15). Having the above optimal reaction conditions in hand, we then explored the scope of the coupling reactions of various 3,4-dihydroisoquinolines with their corresponding partners (Figure 2). All of the experiments were carried out in one hour to furnish the target molecules mostly in moderate-to-good yields and with significant diastereomeric ratios.

The addition reactions of γ-mercapto-α,β-unsaturated methyl ester (**18a**), ethyl ester (**18b**) and phenyl ester (**18c**) with various 3,4-dihydroisoquinolines provided products **20aa**-**20ea**, products **20fb**-**20jb** and products **20kc**-**20nc**, respectively. Furthermore, some 3,4-dihydroisoquinoline derivatives also reacted with γ-mercapto-α,β-unsaturated amide **19** to give the corresponding products (**21a**–**21d**) in moderate-to-good yields.

Presumably, in compounds **18** and **19**, the -SH group is a better nucleophile than the -OH group in compound **11** (Figure 3) to attack the C=N moiety in compound **4**. However, the yields of the products given by the reactions of unsaturated amides were relatively lower than those of unsaturated esters. It might be concluded that the strong electron-donating effect of nitrogen atom to the carbonyl group could reduce the reactivity of the aza-Michael reaction. In addition, it was found that the reactions of the substrates (**4**) bearing MeO- group gave products in relatively low yields. Evidently, the strong electron-donating property of the methoxy group on the benzene ring might reduce the electron demanding property of C=N moiety at the beginning of reaction.

A possible mechanism for the formation of the title compounds is illustrated in Figure 3. At first, the nucleophilic addition of **18** to 3,4-tetrahydroisoquinolines **4** gave structure **A**. Structure **B** was furnished after the intramolecular aza-Michael addition reaction proceeded in **A**. Then, an immediate intramolecular proton transfer occurred to **B** to provide enol **C**, which underwent tautomerism to afford the target molecule **20**. The diastereoselectivities in these reactions could be controlled by the steric effect during the tautomerism. It is also noteworthy that the diastereoselectivities are considerably lower than those observed for the formation of **3** (Figure 1) [10,11]. Since these reactions were carried out at reflux, we assume that the high temperature and high reactivity of the –SH group might affect the diastereoselectivities.

## 3. Conclusions

In conclusion, various derivatives of 3-substituted thiazolo[2,3-α]isoquinolines were synthesized. The crucial step was the addition reaction of 3,4-dihydroisoquinolines with γ-mercapto-α,β-unsaturated methyl esters or α,β-unsaturated amide. This reaction gave the target molecules with good yields and significant diastereomeric ratios. Further applications of the key reaction and final products are under investigation.

## Data Availability

No applicable.

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
