# Peer review of "Facile Synthesis of 3-Substituted Thiazolo[2,3-α]tetrahydroisoquinolines"

_molecules, 2021, doi:10.3390/molecules26206126_

Round 1
Reviewer 1 Report
This manuscript describes successful synthesis of thiazolo[2,3-a]tetrahydroisoquinolines as a extension of the authors’ previous work regarding the synthesis of oxazolotetrahydroisoquinolines. I think this paper is publishable in Molecules after the following comments are taken into account.
Why does 11 not react with 4, while 5, 18, and 19 do react? The authors should mention possible reasons.
Please discuss the origin of diastereoselectivity in the present reactions.
The diastereoselectivities of the present reactions are considerably lower than those observed for the formation of 3 (ref. 6). What is the reason?
Lines 9, 23, 31, and 100, "tetrahydroisoquinoline": Does this read "dihydroisoquinoline"?
Lines 11 and 50: The term "green conditions" is not suitable for the present reactions since they were carried out in the presence of added excess acid.
Line 17: The compound name "sasolidine" is not well recognized. The molecular structure or other information should be indicated.
[2,3-a] is written with italic "a", non-italic "a", and Greek letter alpha in different lines. Please use the same letter throughout.
Supporting Information, lines 80 and 82: Wrong compound numbers 192, 208, and 212.
Author Response
Point 1: Why does 11 not react with 4, while 5, 18, and 19 do react? The authors should mention possible reasons.
Response 1: In compound 11, the strong electron-donating effect of -OR2 group to the carbonyl part on the ester moiety might reduce the reactivity of aza-Michael reaction, while there is no -OR2 group in compound 5 (please see the end of the first paragraph in the revised manuscript). On the other hand, in compounds 18, and 19, the -SH group is presumably a better nucleophile than -OH group to attack the C=N moiety in compound 4. (please see the end of the "Discussion" in the revised manuscript).
Point 2: Please discuss the origin of diastereoselectivity in the present reactions.
Response 2: The diastereoselectivity in these reactions could be controlled by the steric effect in the tautomerism (Please see the end of the "Discussion" right above Figure 2).
Point 3: The diastereoselectivities of the presentreactions are considerably lower than those observed for the formation of 3 (ref. 6). what is the reason?
Response 3: The reason is now described in the final paragraph of "Discussion".
Point 4: Lines 9, 23, 31 and 100, "tetrahydroisoquiniline": Does this read "dihydroisoquiniline"?
Response 4: Yes, they are now all corrected.
Point 5: Lines 11 and 50: The term "green conditions" is not suitable for the present reactions since they were carried out in the presence of added excess acid.
Response 5: The term "green conditions" is changed to "appropriate conditions" every where.
Point 6: Line17: The compound name "sasolidine" is not well recognized. The molecular structure or information should be indicated.
Response 6: It should be "salsolidine", an MAOA inhibitor. Please see line17 of the revised manuscript.
Point 7: [2,3-a] is written with italic "a", non-italic"a"and Greek letter alpha in different lines. Please use the same letter throughout.
Response 7: Now, italic "a" is used throughout.
Point 8: Supporting Information, lines 80 and 82: Wrong compound numbers 192, 208, and 212.
Response 8: Compound numbers are now all corrected.
We thank reviewer 1 for the comments and suggestions.

Reviewer 2 Report
The paper is quite confusing, both scientifically and organizationally.
On the other hand, the introduction is very poor in terms of framing the theme
and it is not clear what is the introduction and what is the results/discussion.
As it is, I do not recommend publishing the paper.
I recommend that the authors reorganize the paper in order to make it more
noticeable and reformulate the introduction.
Author Response
Point 1: The introduction is very poor in terms of framing the theme and it is not clear what is the introduction and what is the results/discussion.
Response 1: The content of "Introduction" is revised. (Please see the end of the first paragraph right above Figure 1). The title "Results and Discussion" is added to the space below Scheme 4 and the content of "Results and Discussion" is revised. (Please see the first paragraph of the revised content of it).
Point 2: I recommend that the authors reorganize the paper in order to make it more noticeable and reformulate the introduction.
Response 2: The content of "Introduction" is reformulated and that of "Results and Discussion" reorganized.
We thank Reviewer 2 for the comments and suggestions.

Reviewer 3 Report
In this manuscript authors introduced a new method for the synthesis of Thiazolo [2,3- a,] isoquinoline derivatives. Mechanism of the major reaction was also illustrated.
Below are my questions and comments on the manuscript:
- The current paper is not well organized, it has only two sections: Introduction and Conclusion. Though the authors provided relevant information, but they are not categorized into sections which makes manuscript hard to read. Manuscript is missing key sections such as Results, Discussion, Materials and methods etc.
- Authors claimed that the reactions were conducted “under green conditions” which seems not true as in reality the reactions were carried in presence of EtoAC as a solvent and AcOH. Authors should clearly explain on what basis they are claiming that the reactions are conducted under green conditions.
- From page 5, table 1, entry 6, author tried the reaction with THF under room temperature and got 69% yield whereas under similar conditions with EtOAc yielded 70% of product. Before selecting entry 15 as an optimized condition, Did the author try a reaction with THF under reflux condition? If so, what is the yield?
- In page 6, table 2, when the reaction was carried in presence of unsaturated amides the yields are low. For example: low yield in 21b, 21c and mainly in 21d. Any interesting side products isolated in these reactions? Author should provide necessary information and discuss the factors that are responsible for low yields.
- From page 6, table 2, it is clearly evident that author conducted fewer experiments with unsaturated amides. Any reason behind this low substrate scope in the case of amides?
- From the results provided in table-2, it is clearly evident that when the reactions are carried out with substrates having MeO- group the yields are low. Any electronic factors effecting the yields? Author needs to discuss this.
- Some of the organic structures (especially structures of compounds 18 & 19) were not properly drawn throughout the paper. Organic structures should be drawn in a way that they represent correct bond angles.
- In the supporting information, page 2, line 41, product structure was wrong (see attached review document for comment).
- In the supporting information, page 3, line 78, structure of compound 18 or 19 is incorrect, it should not be a ketone (see attached review document for more comments on this).
- In the supporting information, page 3, lines 76 - 78, Compound numbers are not matching with the numbers given in the main manuscript.
- In the supporting information, page 3, lines 80 -82, Compound numbers are not matching with the numbers given in the main manuscript (see attached review document for this comment).
For more other comments, please see the attached review document.

Author Response
Point 1: The current paper is not well organized, it has only two sections: Introduction and Conclusion. Though the authors provided relevant information, but they are not categorized into sections which makes manuscript hard to read. Manuscript is missing key sections such as Results, Discussion, Materials and methods etc.
Response 1: The title "Results and Discussion" is added to the space below Scheme 4 and the content of "Results and Discussion" is revised. (Please see the first paragraph of the revised content of it). Materials and methods are shown in "Supporting Information".
Point 2: Authors claimed that the reactions were conducted “under green conditions” which seems not true as in reality the reactions were carried in presence of EtoAC as a solvent and AcOH. Authors should clearly explain on what basis they are claiming that the reactions are conducted under green conditions.
Response 2: The term "green conditions” is now changed to "appropriate conditions" where it is applicable.
Point 3: From page 5, table 1, entry 6, author tried the reaction with THF under room temperature and got 69% yield whereas under similar conditions with EtOAc yielded 70% of product. Before selecting entry 15 as an optimized condition, Did the author try a reaction with THF under reflux condition? If so, what is the yield?
Response 3: Since ether acetate is a less toxic solvent than THF, the authors did not try a reaction in THF under reflux condition.
Point 4: In page 6, table 2, when the reaction was carried in presence of unsaturated amides the yields are low. For example: low yield in 21b, 21c and mainly in 21d. Any interesting side products isolated in these reactions? Author should provide necessary information and discuss the factors that are responsible for low yields.
Response 4: There were no side products observed from the reaction of unsaturated amides. Since the yields of the products given by the reactions of unsaturated amides were relatively lower than those of unsaturated esters, it might be concluded that the strong electron-donating effect of nitrogen atom to the carbonyl group might reduce the reactivity of aza-Michael reaction. (Please see the end of "Results and Discussion" in the revised manuscript).
Point 5: From page 6, table 2, it is clearly evident that author conducted fewer experiments with unsaturated amides. Any reason behind this low substrate scope in the case of amides?
Response 5: Since the starting material other than compound 24 (Scheme 6) is not commercially available, the authors could only prepared compound 19 for the reaction of unsaturated amide.
Point 6: From the results provided in table-2, it is clearly evident that when the reactions are carried out with substrates having MeO- group the yields are low. Any electronic factors effecting the yields? Author needs to discuss this.\
Response 6: Yes, we discuss the electronic factors at the end of "Results and Discussion" in the revised manuscript.
Point 7: Some of the organic structures (especially structures of compounds 18 & 19) were not properly drawn throughout the paper. Organic structures should be drawn in a way that they represent correct bond angles.
Response 7: Yes, we revised those organic structures and redrawn them.
Point 8: In the supporting information, page 2, line 41, product structure was wrong (see attached review document for comment).
Response 8: It is corrected,now.
Point 9: In the supporting information, page 3, line 78, structure of compound 18 or 19 is incorrect, it should not be a ketone (see attached review document for more comments on this).
Response 9: It is revised,now.
Point 10: In the supporting information, page 3, lines 76 - 78, Compound numbers are not matching with the numbers given in the main manuscript.
Response 10: It is revised ,now.
Point 11: In the supporting information, page 3, lines 80 -82, Compound numbers are not matching with the numbers given in the main manuscript (see attached review document for this comment).
Response: 11: It is revised, now.
Point 12: For more other comments, please see the attached review document.
Response 12: Yes, we thank reviewer 3 for the kind comments and suggestions.

Round 2
Reviewer 2 Report
The authors perfermed all the changes required. I consider that this paper can be published as it is.
Reviewer 3 Report
Author has addressed almost all the comments that were raised during the first review. I recommend to replace the wording "green conditions" with "appropriate conditions" in line 25 and 56. The manuscript can be published after this minor change.